RESEARCH　　　　　　　　　　　　　　　　　　　　　　　　　　　　　　　Open Access

# Wheat chromatin architecture is organized in genome territories and transcription factories

Lorenzo Concia[1], Alaguraj Veluchamy[2], Juan S. Ramirez-Prado[1], Azahara Martin-Ramirez[3], Ying Huang[1], Magali Perez[1], Severine Domenichini[1], Natalia Y. Rodriguez Granados[1], Soonkap Kim[2], Thomas Blein[1], Susan Duncan[4], Clement Pichot[1], Deborah Manza-Mianza[1], Caroline Juery[5], Etienne Paux[5], Graham Moore[3], Heribert Hirt[1,2], Catherine Bergounioux[1], Martin Crespi[1], Magdy M. Mahfouz[2], Abdelhafid Bendahmane[1], Chang Liu[6], Anthony Hall[4], Cécile Raynaud[1], David Latrasse[1] and Moussa Benhamed[1,7]*

## Abstract

**Background:** Polyploidy is ubiquitous in eukaryotic plant and fungal lineages, and it leads to the co-existence of several copies of similar or related genomes in one nucleus. In plants, polyploidy is considered a major factor in successful domestication. However, polyploidy challenges chromosome folding architecture in the nucleus to establish functional structures.

**Results:** We examine the hexaploid wheat nuclear architecture by integrating RNA-seq, ChIP-seq, ATAC-seq, Hi-C, and Hi-ChIP data. Our results highlight the presence of three levels of large-scale spatial organization: the arrangement into genome territories, the diametrical separation between facultative and constitutive heterochromatin, and the organization of RNA polymerase II around transcription factories. We demonstrate the micro-compartmentalization of transcriptionally active genes determined by physical interactions between genes with specific euchromatic histone modifications. Both intra- and interchromosomal RNA polymerase-associated contacts involve multiple genes displaying similar expression levels.

**Conclusions:** Our results provide new insights into the physical chromosome organization of a polyploid genome, as well as on the relationship between epigenetic marks and chromosome conformation to determine a 3D spatial organization of gene expression, a key factor governing gene transcription in polyploids.

**Keywords:** Hi-C, Hi-ChIP, DNA loops, Transcription factories, Genome territories

* Correspondence: moussa.benhamed@u-psud.fr
[1]Institute of Plant Sciences Paris of Saclay (IPS2), UMR 9213/UMR1403, CNRS, INRA, Orsay, France
[7]Institut Universitaire de France (IUF), Paris, France
Full list of author information is available at the end of the article

## Background

In eukaryotes, nuclear DNA is folded into chromatin, a tightly packed high-order structure whose main components are DNA and histone proteins. Chromatin conformation determines the accessibility of the double helix to the transcriptional and replication machineries [1]. It is tightly regulated in both time and space by highly conserved mechanisms such as DNA methylation, histone post-translational modifications, alterations in histone-DNA interactions, and incorporation of histone variants by chromatin remodelers, as well as long non-coding RNA (lncRNA)- and small RNA (sRNA)-related pathways [2–4].

In light of the strong influence of chromatin conformation on gene expression, the idea of the genome as a linear sequence of nucleotides has been replaced by a dynamic three-dimensional (3D) architecture [5] in which structural elements such as "loops", "domains", "territories", and "factories" are functional components controlling the physical interactions between promoters and distant regulatory elements [6]. Accordingly, the nucleus is thought to operate as an integrated regulatory network [7].

Recent development of new methods for the analysis of the genome-wide 3D spatial structure of chromatin, such as Hi-C, Hi-ChIP, and ChIA-PET, has made it possible to unveil small- and large-scale genome topology in various cell types of metazoan organisms, notably in mammals [8]. These studies revealed the existence of megabase-long chromatin compartments comprising either open, active chromatin (A compartment) or closed, inactive chromatin (B compartment). Chromatin conformation capture techniques also allowed the description of topologically associating domains (TADs), large chromatin domains (800 kb average length in mammals) bringing together contiguous sequences of DNA. Genes that belong to the same TAD display similar dynamics of expression, suggesting that physical association is functionally relevant to gene expression control.

The basic organization of plant genomes differs from that in animals [9–14]. For instance, plants do not display an apparent A/B compartmentalization as a predominant genome folding feature [14, 15].

Moreover, although TAD-like domains have been identified in maize, tomato, sorghum, foxtail millet, and rice [15], no canonical insulator proteins have been described in plants, challenging the classification of these structures [12]. This may be due to the fact that the plant genome structure has a particular transposable element content and location relative to genes and opens the question of the functional conservation of genome folding in eukaryotes.

Another particular feature of plant genomes is the high occurrence of polyploidy, which leads to the co-existence of several copies of similar genomes deriving from the same (autopolyploid) or related (allopolyploidy) species in the same nucleus. How polyploidy affects chromosome folding architecture and functional organization remains to be elucidated. Established polyploids often have higher fitness attributes [16], and polyploidy is considered a major factor in successful plant domestication [17] as demonstrated by its widespread presence among cultivated crops, both autopolyploids (e.g., potato [*Solanum tuberosum*], alfalfa [*Medicago truncatula*], banana [*Musa acuminata*], and watermelon [*Citrullus lanatus*]) and allopolyploids (e.g., canola, strawberry, cotton, coffee, sugarcane, and wheat) [18].

Polyploidization events trigger extensive epigenetic and transcriptional alteration of the duplicated or merged genomes, accompanied by small- and large-scale conformational changes [18–23]. Such rearrangements are thought to have facilitated the domestication of polyploid crops by enhancing their adaptive plasticity [17]. The characterization of 3D chromosome topology in polyploids crops may help better understand the degree to which spatial organization contributes to polyploidy success.

Modern hexaploid wheat (*Triticum aestivum* L.; $2n = 6x = 42$) is a particularly interesting model for analyzing chromosome topology, because it is the product of two rounds of interspecific hybridization that occurred at different evolutionary times (Fig. 1a). The first hybridization event occurred 0.36 to 0.50 million years ago and produced a tetraploid species, *Triticum turgidum* (AABB) [24–27]. This hybridization involved *Triticum urartu* (donor of the AA genome) and an unknown species related to *Aegilops speltoides* (BB genome) [28]. Indeed *Ae. speltoides* cannot be considered as the exclusive donor of this genome, but the wheat B genome might rather have a polyphyletic origin with multiple ancestors involved, among which *Ae. speltoides*. A second hybridization event between *T. turgidum* (AABB) and the diploid species *Aegilops tauschii* (DD genome) gave rise to a hexaploid wheat (AABBDD), the ancestor of the modern bread wheat, about 10,000 years ago [25, 29]. Since the three ancestors are closely related species descended from a common progenitor, three distinct but highly syntenic subgenomes can be identified (AA, BB, and DD) [30]. Compared to tetraploid wheat, modern hexaploid wheat possesses several agricultural advantages, such as increased environmental adaptability, tolerance to abiotic stresses (including salinity, acid pH, and cold), and increased resistance to several pathogens, factors that contribute to its success as a crop [31]. Although the genetic determinants of wheat yield and quality have been extensively investigated [32, 33] and a fully annotated reference genome was recently generated together with tissue-specific and developmental transcriptomic co-expression networks [34], the influence of

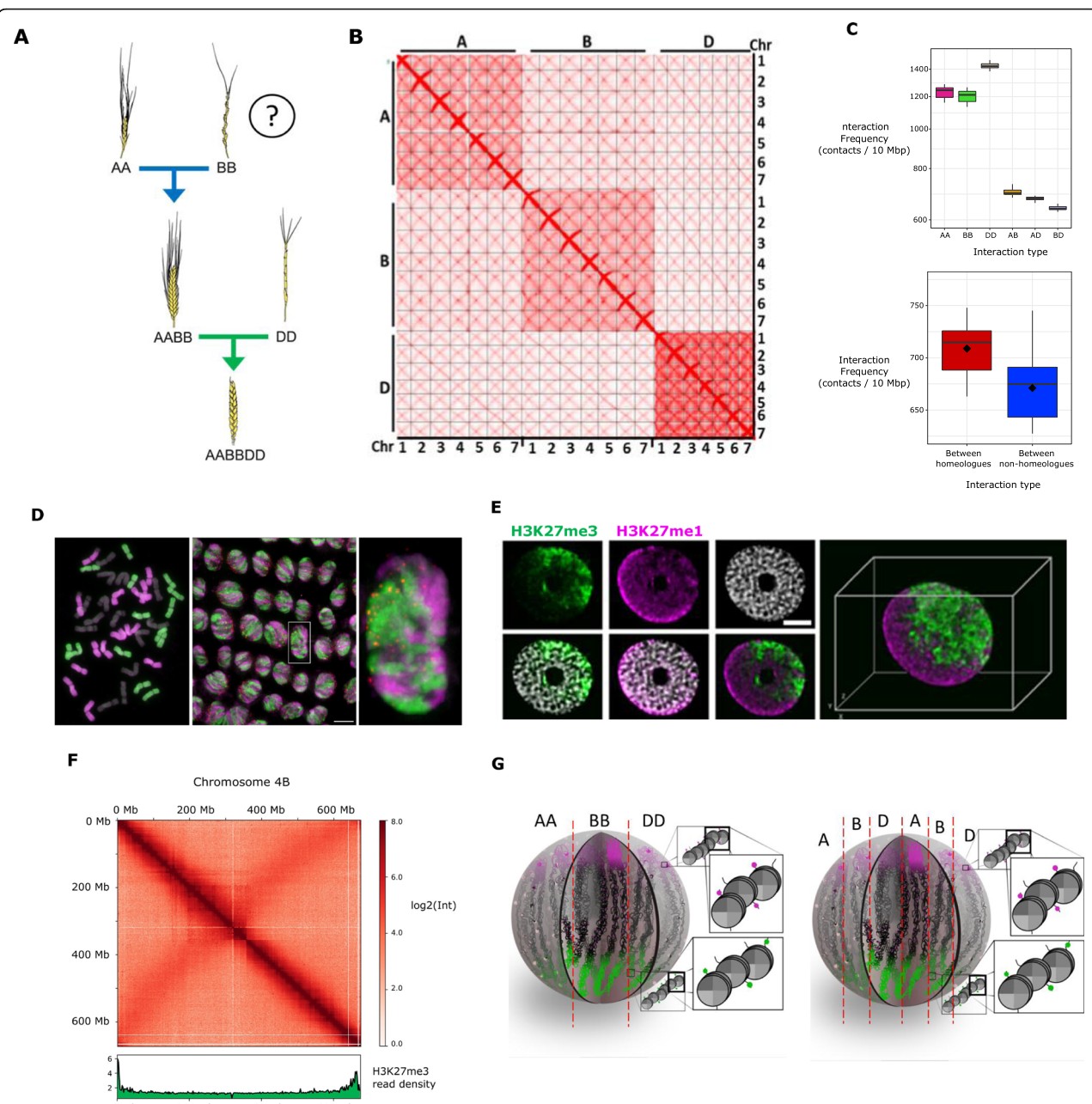

**Fig. 1** Large-scale chromatin architecture analysis of hexaploid wheat. **a** Schematic representation of the relationships between wheat genomes, showing the polyploidization history of hexaploid wheat. **b** Hi-C contact matrix of the hexaploid wheat genome. **c** Box plots representing the distribution of the median interaction frequency between 10-Mb bins for each combination of subgenomes (upper panel) and between homoeologous and non-homoeologous chromosomes of different subgenomes (bottom panel). **d** Root meristematic cells of *T. aestivum* cv. Chinese Spring labeled by GISH. The A genome is labeled in magenta, the D genome is labeled in green, and the B genome is not labeled and thus appears in gray; telomeres are labeled in red. (Left panel) metaphase cells showing 14 A chromosomes, 14 B chromosomes, and 14 D chromosomes. (Middle panel) interphase cells. (Right panel) zoom-in of the interphase nucleus indicated by the white box in the middle panel. Scale bar represents 10 μm. **e** Immunofluorescence detection of H3K27me3 (green) and H3K27me1 (purple) in the isolated nucleus. **f** Integration of H3K27me3 and Hi-C at the chromosomal level. The heatmap of intrachromosomal interaction frequency of chromosome 4 B (in red, upper panel) is presented together with the read density of H3K27me3 ChIP-seq (in green, lower panel). The genomic coordinates are indicated on the left side of the heatmap for Hi-C and under the read density plot for H3K27me3. The color bar on the right side of the heatmap shows the interaction frequency scale. **g** Alternative models of large-scale chromatin architecture in the nucleus of wheat

chromatin organization on the expression of key traits of agricultural interest is still poorly understood.

In this study, we present an analysis of hexaploid wheat nuclear architecture by the integration of Hi-C, ChIP-seq, RNA-seq, and Hi-ChIP data. Our results highlight at least three levels of DNA spatial organization: (i) an arrangement into genome territories, (ii) a diametrical separation between facultative and constitutive heterochromatin, and (iii) the organization of active chromatin DNA around transcription factories, established through micro-compartmentalization and physical interactions of these genes carrying euchromatic histone modifications.

## Results

### Wheat chromatin is organized in genome territories

To gain insight into wheat chromatin architecture, we used Hi-C, a genome-wide chromatin conformation capture method that detects DNA-DNA physical interactions [35] to examine shoot cells of 14-day-old seedlings of *T. aestivum* cv. Chinese Spring. Analysis of the whole-genome interaction matrix (Fig. 1b) revealed three hierarchical layers of chromosome interactions, from strongest to lowest: (i) within chromosomes, (ii) between chromosomes of the same subgenome, and (iii) between chromosomes of different subgenomes. This organization indicates a non-random spatial distribution of the three subgenomes that could mirror the presence of functional "genome territories." We quantified these differences by plotting the distribution of median interaction frequency between 10-Mb bins for intrachromosomal interactions, interchromosomal interactions within each subgenome, and interchromosomal interactions between subgenomes (Additional file 1: Supplemental Figure S1) and between pairs of subgenomes (Fig. 1c, middle panel). In addition, we observed that the A and B genomes interact more frequently than A and D or B and D. We independently confirmed the presence of subgenome-specific territories using a genomic in situ hybridization (GISH) experiment on root meristematic cells (Fig. 1d).

Then, we examined the intergenomic interaction frequency between homeolog and non-homeolog chromosomes of different subgenomes (Fig. 1c, lower panel), finding that homeologs interact more frequently than non-homeolog chromosomes (Mann-Whitney p = 2.18e −06; Cliff's Delta effect size = 0.65). This imbalance was not anticipated and could be the macroscopic manifestation of specific contacts between homeolog genes, whose expression has been reported to be often coordinated in space and time [36]. Root cells presented the same chromosome interaction pattern (Additional file 1: Supplemental Figure S1).

To verify whether other polyploid plants share the same large-scale nuclear organization, we carried out a Hi-C assay on 14-day-old seedlings of rapeseed (*Brassica napus*

L.) (Additional file 1: Supplemental Figure S2). The experiment revealed again a three-layer hierarchy of chromosome interactions identical to wheat, suggesting that this organization is a general feature of polyploid plants.

To further analyze wheat chromatin organization, we performed nucleus immunostaining with antibodies directed against different histone marks. We first labeled constitutive and facultative heterochromatin using antibodies recognizing H3K27me1 and H3K27me3, respectively (Fig. 1e). H3K27me3 plays a critical role in the epigenetic silencing of developmentally or stress-regulated genes in the context of facultative (reversible) heterochromatin, while H3K27me1 is required for the formation and maintenance of constitutive heterochromatin, and thus participates in the inhibition of TE expression [11, 37].

We observed an opposite and polarized distribution of these two marks throughout the nucleus, which suggests the presence of subnuclear domains with different types of chromatin. A second immunostaining with an antibody recognizing H3K9me2, another histone modification associated with constitutive heterochromatin [38, 39], confirmed this result (Additional file 1: Supplemental Figure S3, upper panel). We then examined active chromatin by immunostaining with an antibody directed against H3K36me3, a euchromatic mark linked to active transcription [40, 41] (Additional file 1: Supplemental Figure S3, lower panel). Contrary to the previous distribution, H3K36me3 was localized in sharp foci scattered across the whole nucleus, suggesting a physical proximity of clusters of transcriptionally active regions.

To investigate the relationship between chromatin 3D folding and the observed polarization of facultative and constitutive heterochromatin, we integrated H3K27me3 ChIP-seq data and Hi-C data at the chromosome-wide level. Single-chromosome interaction matrixes showed a strong signal over the main diagonals, consistent with distance-dependent decay of interaction frequency, in agreement with the previous literature [35] (Fig. 1b, f; Additional file 1: Supplemental Figure S4). The second strong signal was detectable in an antidiagonal direction due to extensive interactions between the two arms of each chromosome. This conformation is consistent with the Rabl organization previously described in wheat [42] and barley [14]. The comparison with H3K27me3 ChIP-seq data (Fig. 1f) revealed that the peaks of read density are located primarily in the subtelomeric regions of each chromosome arm. Globally, this explains how the Rabl configuration in which subtelomeres are juxtaposed, together with the clustering of H3K27me3-labeled genes in subtelomeric regions, produces the polarized distribution of epigenetic marks and supports the existence of functional compartmentalization within the nucleus of bread wheat.

## The wheat genome displays intergenic condensed spacer (ICONS) folding structures

Topologically associating domains (TADs) are defined as genomic regions containing sequences that interact more frequently with others in the same TAD than with those in different TADs [43, 44]. In metazoans, the boundaries of TADs are enriched in specific proteins involved in their maintenance, such as the CTCF (CCCTC binding factor) transcriptional repressor and cohesin [45–47]. The functional conservation in plants of these folding structures remains unclear. We therefore wanted to investigate the possible presence of folding domains in wheat. After normalizing the Hi-C interaction matrixes for technical and biological biases that could alter the number of reads aligning on each fragment, such as GC content, distance between restriction sites, and mappability with iterative correction and eigenvector decomposition (ICE) (Imakaev et al. 2012), we applied the "insulation index" technique. The insulation index of a genomic region is calculated as the average frequency of interaction with the neighboring regions within a predefined distance [48]. We found 32, 299 folding domains, covering 51% of the genome with an average size of about 225 kbp (Additional file 1: Supplemental Figure S5).

A prominent feature of the insulation index was the presence of genes, both protein-coding and non-coding such as microRNA precursors or long non-coding RNA, in correspondence of local minima and in particular at the boundaries of folding domains (Fig. 2b–e); we subsequently named these structures ICONS, for intergenic condensed spacers. Examining the distribution of the active chromatin marks H3K9ac and H3K36me3 and of the facultative heterochromatin mark H3K27me3 over ICONS, we found all three histone marks are enriched at the boundaries, as predicted by the presence of genes at those locations. In contrast, both transposable elements and DNA methylation on CpG and CHG contexts were depleted at the boundaries and enriched within ICONS (Additional file 1: Supplemental Figure S6). The result of an ATAC-seq experiment (assay for transposase-accessible chromatin using sequencing) [49, 50] revealed that chromatin is highly accessible at the ICONS boundaries and less within ICONS (Additional file 1: Supplemental Figure S6). Taken together, these features indicate that the wheat genome presents distinctive folding domains (ICONS) rather than canonical topologically associating domains (TADs).

To analyze more in-depth the relationship between histone modifications, chromatin accessibility, and chromatin interaction, we took into account the average insulation index over protein-coding and non-coding genes by comparing the degree of spatial co-localization through peaks of each histone mark or ATAC-seq (Fig. 3a–c, Additional file 1: Supplemental Figure S6 and S7). We discovered that the insulation index is more

negative for genes overlapping with peaks of the active marks H3K9ac and H3K36me3 than for those without these histone marks, while genes carrying the repressive mark H3K27me3 displayed the opposite behavior. This result highlights that, while genes generally interact weakly with their neighboring sequences, those bearing euchromatic histone marks are especially devoid of physical contact with the surrounding regions. To determine if a quantitative relationship exists between the level of a specific histone modification and the intensity of chromatin physical interactions, we partitioned the genome in ten deciles of insulation index and plotted the median read density of each histone mark over the genes contained in each group (Fig. 3d–f; Additional file 1: Supplemental Figure S7). We observed that the read density of H3K27me3 gradually increased with the insulation index whereas those of H3K9ac and H3K36me3 showed a negative correlation supporting that histone modification may play a major role in chromatin interaction in wheat.

This trend prompted us to examine whether gene expression is affected by local interactions. We first compared insulation indexes over coding or non-coding genes overlapping or non-overlapping with RNA polymerase II ChIP-seq peaks (Fig. 3g and Additional file 1: Supplemental Figure S7) and expressed or non-expressed genes (Fig. 3h). We found that genes bound to RNA polymerase II and producing detectable transcripts have on average a more negative insulation index. Consistently with the distribution of histone marks, the frequency of interaction of transcribed genes with flanking regions was inversely proportional to their expression level given in "transcripts per million" (TPM) (Fig. 3i). We then considered the read density of RNA polymerase II ChIP-seq over each of the 10 insulation quantiles described above and found that genes characterized by a less negative insulation index are enriched in RNA polymerase II ChIP-seq (Fig. 3j and Additional file 1: Supplemental Figure S7). According to these results, the frequency of interaction between genes and the surrounding regions correlates with both their histone marks and their expression level, suggesting that local architecture is influenced by epigenetic mechanisms acting in transcriptional regulation.

## Chromatin loops define local-scale functional units of wheat genome architecture

The visualization of the interaction matrix revealed the widespread presence of interaction hotspots between genomic bins containing genes, a strong indication of gene-to-gene loops (GGLs) (Fig. 2b, d, Fig. 4a). To confirm the presence of such structures, using the software HOMER (see the "Materials and methods" section), we identified 293,044 loops in shoots (see the "Materials

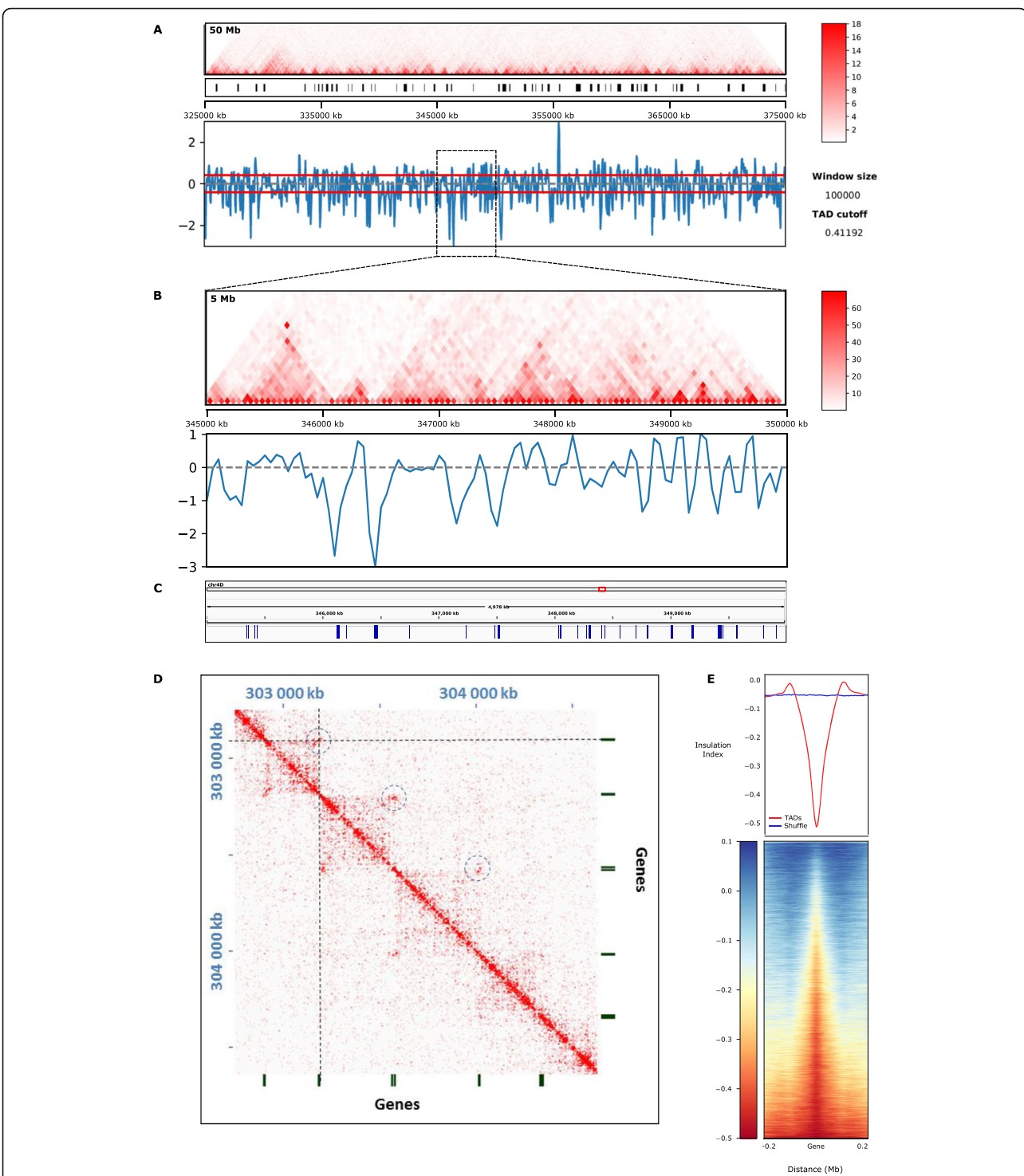

**Fig. 2** The wheat genome displays ICONS. **a** TADtool analysis of a 50-Mb region of chromosome 4D (Chr4D:325000000-375000000). Triangle heatmap of Hi-C interaction frequency (top panel), positions of the identified TADs indicated by black bars (middle panel), and plot of the insulation index (bottom panel). **b** Zoom-in of the 5-Mb region of Chr4D:345000000-350000000 in **a**, showing a triangle heatmap of Hi-C interaction frequency (top panel) and insulation index (bottom panel). **c** Position of genes along the 5-Mb region shown in **b**. **d** 2D heatmap showing the interaction frequency in a region (chr3B-1000000–5000000) of the Hi-C map. The dashed circles highlight the hotspots of interaction. Genes are represented by black bars. **e** Level of insulation index on genes. Median insulation index over genes (red line) and random genomic intervals (blue line) (top panel). Heatmap of insulation index over genes (bottom panel)

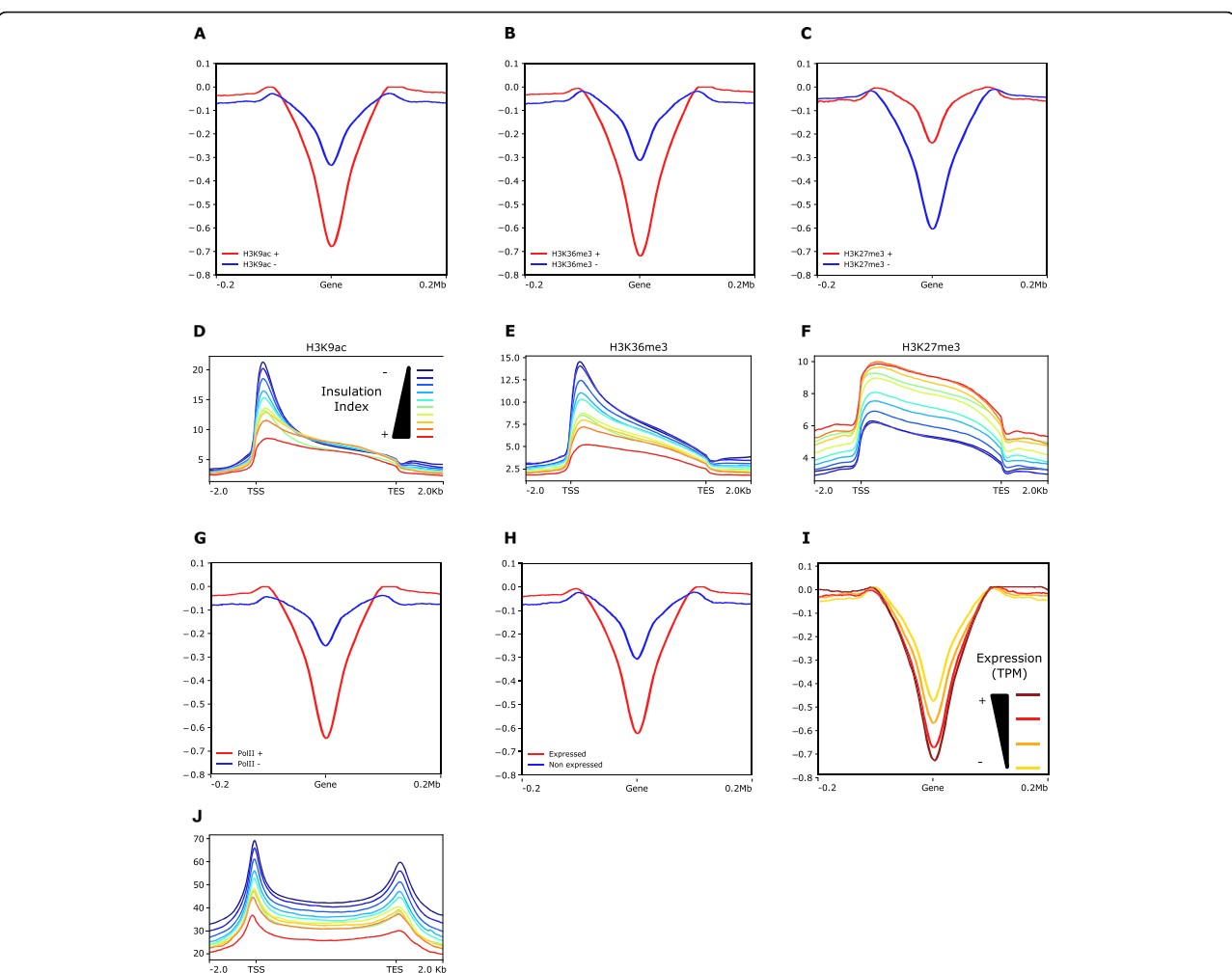

**Fig. 3** Relationship between insulation index, histone marks, and gene expression. **a–c** Plots displaying the insulation index over genes marked (red line) or not marked (blue line) by H3K9ac (**a**), H3K36me3 (**b**), and H3K27me3 (**c**). **d–f** Metaplots showing the normalized ChIP-seq read density of H3K9ac (**d**), H3K36me3 (**e**), and H3K27me3 (**f**) median enrichment over the genes categorized from low (first quantile, blue line) to high insulation index (tenth quantile, red line). **g** Plot displaying the insulation index over genes bound to RNAPII (red line) or not bound (blue line). **h** Plot displaying the insulation index over genes expressed (red line) or not expressed (blue line). **i** Plot displaying the insulation index over expressed genes categorized from low expression level (first quartile yellow line) to high expression level (fourth quartile brown line). **j** Metaplots showing the normalized RNAPII ChIP-seq read density over genes categorized from low (first quantile, blue line) to high (tenth quantile, red line) insulation index. Normalized ChIP-seq read densities along the gene and 2-kb region flanking the TSS or the TES are shown

and methods" section), of which 35.6% involved at least one gene and 13.4% between two gene-containing bins. Overall, 47,763 genes (28.8% of all genes) were associated with one or more GGLs. The presence of gene-to-gene loops (GGLs) was independently confirmed with a 3C-qPCR assay for four gene pairs spanning a distance between 200 and 400 kb (Additional file 1: Supplemental Figure S8) [51–53].

Aiming to understand the role of GGLs in the regulation of gene expression, we examined the epigenetic marks and transcriptional status of gene pairs associated with GGLs (Fig. 4a–e). To facilitate the analysis, we compiled a table integrating GGL position, size, strength, and presence or absence of functional marks such as

ChIP-seq peaks of H3K9ac, H3K27me3, H3K36me3, and RNA polymerase II on the genes of interest, plus their transcriptional status (expressed or not) and expression level in TPM (Additional file 2: Table S1). We counted the frequencies of each feature on genes involved in GGLs and crossed the frequencies of the first and the second gene (see the "Materials and methods" section). For all combinations of features, we then generated a $2 \times 2$ contingency table on which we calculated the odds ratios (OR) and effect size (Cramer's *V*), two statistics that express the strength of the association between two categorical variables [54] (Additional file 3: Table S2).

A heatmap with the values of $\log_2$(odds ratio) is shown in Fig. 4b. A positive value of $\log_2$(odds ratio) means a

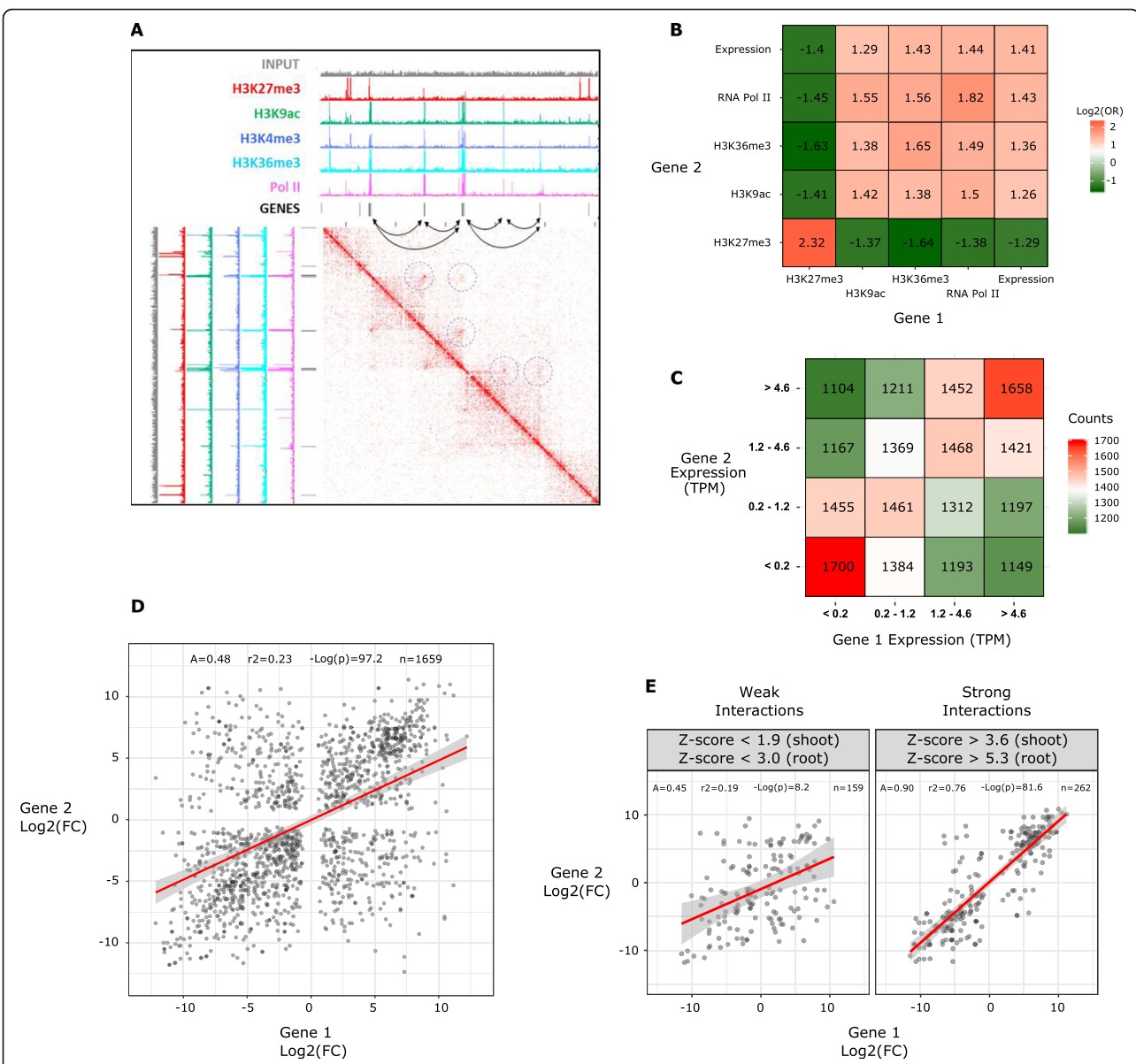

**Fig. 4** Chromatin loops define local-scale functional units of wheat genome architecture. **a** Integration of Hi-C and ChIP-seq data. A 2D heatmap with the interaction frequency in region chr3B-334500000-337000000 is presented. Black bars represent genes. Dashed circles represent the chromatin loop interactions. **b** Heatmap presenting the logarithm of odds ratios of all combinations of features of interacting genes (see the "`Results`" section). Positive log$_2$(odds ratio) indicates enrichment and negative indicates depletion. **c** Heatmap presenting the gene expression levels of gene-to-gene loop (GGL) gene pairs. Within each gene pair, each gene was classified into a quartile of expression, and then the number of gene pairs with each combination of quartiles was counted. **d** Scatterplot of log$_2$(shoot/root fold change) for pairs of interacting genes associated with a GGL conserved in shoots and roots. Loops (significant interactions) were identified from Hi-C data. **e** Scatterplots of log$_2$(shoot/root fold change) for two subsets of loops: weak (first quartile of Z-score) and strong (last quartile of Z-score)

positive association between the two marks: when one mark is present on the first gene, the other marks tend to be present on the second gene.

This approach revealed that GGLs occur more frequently between genes with concordant transcriptional status, whether transcribed or not [log$_2$(odds ratio) = 1.413, Cramer's V = 0.23]. The marks collectively define two main classes of GGLs: the first occurring between expressed genes with active marks (H3K36me3, H3K9ac, and RNA polymerase II) ("active loops") and the second between non-expressed genes bearing the repressive mark H3K27me3 ("repressive loops").

We then asked whether the presence of GGLs is predictive of co-expression between the two genes associated by the loop. To address the question, we considered a subset of GGLs involving only expressed genes (TPM >

0) and split the partner genes into two series of quartiles according to their TPM. The count of genes falling into each combination of quartiles was used to generate a 2D heatmap that unveiled a strong preference for the establishment of GGLs between genes with similar expression levels (Fig. 4c), implying a correlation between transcriptionally active gene pairs and their 3D spatial proximity.

We subsequently wanted to probe whether gene pairs connected in GGLs show similar changes in the expression levels between shoots and roots. Using publicly available gene expression data [55], we identified 21,796 differentially expressed (DE) genes between wheat seedling shoots and roots ($p$ value < 0.01) (see the "Materials and methods" section).

We performed a Hi-C experiment on roots, identified loops with HOMER, and selected those conserved in both organs and containing DE genes (1659 DE-GGLs). We then applied a linear regression to the 1659 gene pairs using the $\log_2$(fold change shoot/root) of the first gene as the predictor and the $\log_2$(fold change shoot/root) of the second gene as the response (Fig. 4d). The regression was highly significant ($F$ test $p$ value = 6.6e −98), and the slope coefficient was positive (0.483), showing that DE genes in physical contact tend to have similar changes in the expression levels between shoots and roots, a behavior consistent with the existence of coregulation mechanisms.

In light of the GGL size distribution (Additional file 1: Supplemental Figure S9), we tested whether the linear distance between interacting genes had an influence on their coregulation and found no differences between short- and long-range interactions (Additional file 1: Supplemental Figure S10).

In addition, we examined whether the correlation between the $\log_2$(fold change) of the two partner genes could be predicted by the "loop strength" of the DE-GGLs, as measured by the $Z$ value assigned by HOMER based on the ratio of observed to expected reads (see the "Materials and methods" section). Similar to the analysis performed for the expression level in TPM (see above), we split the DE-GGLs into two series of quartiles according to the strength in shoots and roots and repeated the analysis for each group (defined by a combination of quartiles) (Fig. 4e and Additional file 1: Supplemental Figure S11). The $p$ value and slope of the regression observed for DE-GGLs were strongly driven by the strongest group in shoots and roots (Fig. 4e). The slope of the regression calculated using only this subset is 0.90, an almost perfect correlation (slope equal to 1). Overall, we observed that GGLs occurred predominantly between genes that share similar epigenetic marks and expression levels. Based on the co-occurrence of these features, we could distinguish two major types of GGLs: active loops and repressive loops. Furthermore, when differentially expressed, the two genes involved in a GGL tend to be coregulated. While invariant to the linear distance between partner genes, the coregulation appears to be more stringent for stronger GGLs. These findings represent strong evidence that the local-scale chromatin architecture plays a role in transcriptional coregulation.

## RNA polymerase II-associated loops organize active chromatin into transcription factories

The results of both Hi-C and RNA polymerase II ChIP-seq (Fig. 4a–e and Fig. 3g, j) raise the hypothesis that the wheat chromatin is organized around transcription factories. We performed immunostaining experiments using an anti-RNA polymerase II antibody and observed RNA polymerase II foci as expected (Fig. 5a). We sought to further validate the presence of transcription factories using the recently developed Hi-ChIP protocol, a method for sensitive and efficient analysis of protein-centric chromosome conformation, with an anti-RNA polymerase II antibody [56]. The density of short reads aligned on the same fragment, commonly referred to as "dangling ends," showed the same strong enrichment over gene bodies as our RNA polymerase II ChIP-seq datasets (Additional file 1: Supplemental Figure S12). The Hi-ChIP datasets confirmed the presence of genomic territories revealed by the Hi-C experiments (Additional file 1: Supplemental Figure S13).

To take full advantage of the higher resolution provided by Hi-ChIP, we used HOMER to identify intrachromosomal RNA polymerase II-associated loops (RALs) (Fig. 5b–d and Fig. 6a–c). We found 27,886 and 15,887 intrachromosomal RALs in shoots and roots, respectively.

We explored the relationship between RALs and differential gene expression with a regression analysis like that carried out for DE-GGLs (see above), obtaining better fitting ($R^2$ = 0.61 versus 0.23) more positive 488 slope (0.77 versus 0.48) (Fig. 5d).

These indicate that the coregulation between genes involved in RNA polymerase II-associated loops is stronger than between genes associated to loops independent from RNA polymerase II, pointing to a role of chromatin loops in the localized action of RNA polymerase II.

To determine whether RNAPII-associated interaction coregulation extends to genes located on different chromosomes, we identified 51,112 and 72,762 interchromosomal interactions in shoots and roots, respectively (Fig. 6). We then tested whether interchromosomal RNAPII-associated interaction, like GGLs, is predictive of co-expression by partitioning them in quantiles according to the expression level of the gene pairs involved (see above). The gene counts in each combination of quantiles (Fig. 6b) confirmed that genes establishing interchromosomal interactions associated to RNAPII tend

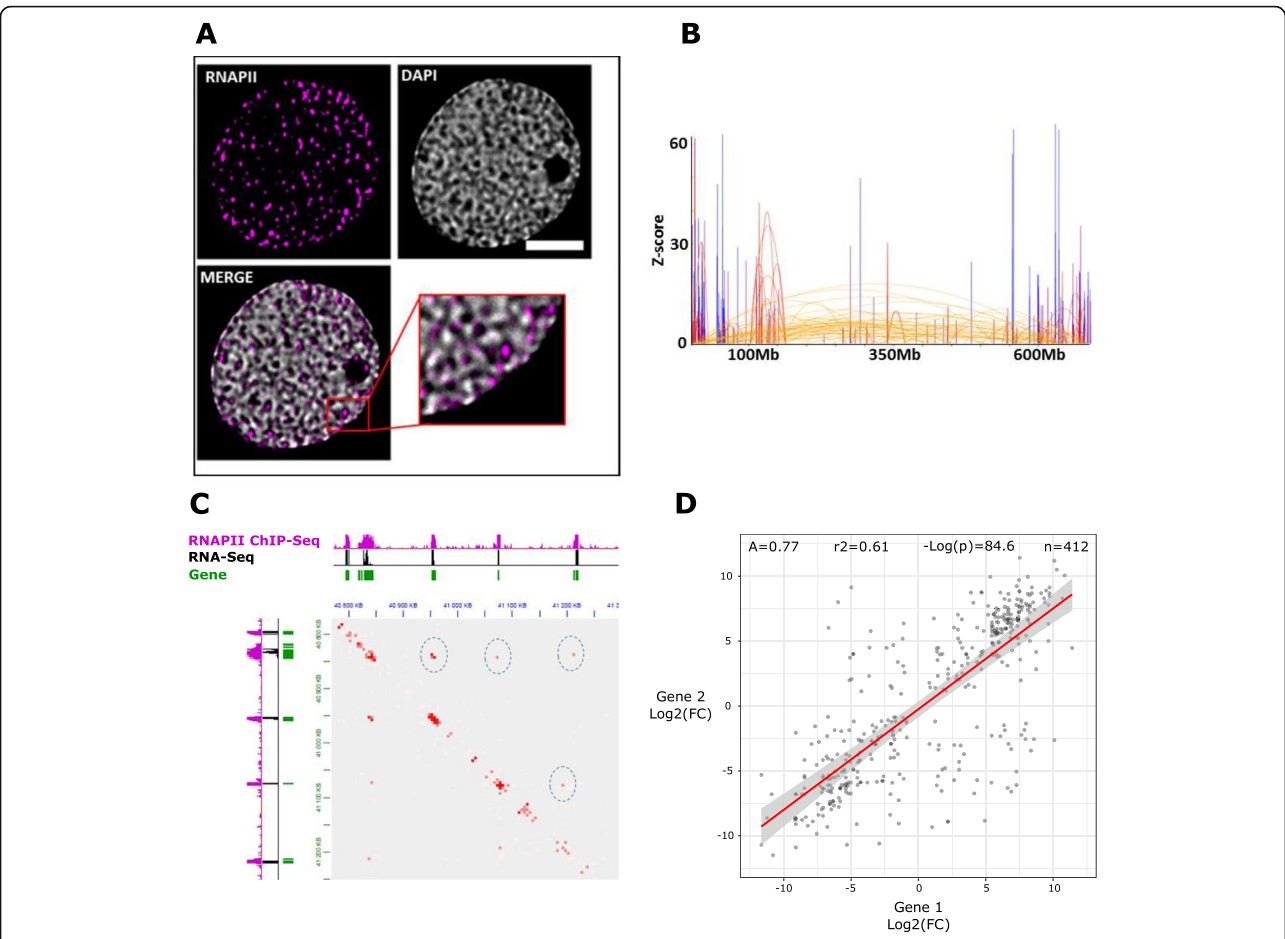

**Fig. 5** Wheat chromatin is organized around transcription factories. **a** Immunofluorescence detection of RNAPII (purple) in the isolated nucleus. **b** Plot of the chromosome 3B RNAPII Hi-ChIP data. The color is associated with a distance range. **c** Integration of RNAPII Hi-ChIP, RNAPII ChIP-seq, and RNA-seq data. A 2D heatmap with the interaction frequency in region chr4D-40800000-41300000 is presented. Green bars represent genes. **d** Scatterplot of log₂(shoot/root fold change) for pairs of interacting genes associated through intrachromosomal RNAPII-associated loops (RALs). Loops (significant interactions) were identified by an RNAPII Hi-ChIP experiment (see the "Results" section). Only the interchromosomal loops conserved between shoots and roots were used in the analysis

to have very similar expression levels. We subsequently asked whether differentially expressed genes interacting through these interchromosomal contacts displayed the same behavior in both shoots and roots. We identified 3947 differentially expressed genes associated with conserved interactions in shoots and roots and repeated the linear regression analysis (Fig. 6c). The outcome confirmed a positive correlation between $\log_2$(fold change) of the gene pairs (regression slope of 0.60), consistent with that observed for RNAPII-associated loops.

We checked the effect of the RNAPII-associated interaction strength on the coregulation of interacting gene pairs by splitting them into two series of quartiles based on their strength in shoots and roots (see above) and found that the interaction strength does not have a major effect (Additional file 1: Supplemental Figure S14).

Finally, we counted the number of partners of each gene involved in RNAPII-associated contacts, finding

that 50% have 4 or more partners and 11% have 10 or more partners (Fig. 6d).

Taken together, (i) the presence of both intra- and interchromosomal RNAPII-associated contacts, (ii) the tendency of these contacts to involve multiple genes, and (iii) the effects of interaction on gene expression and coregulation led us to propose a model for *T. aestivum* in which RNA polymerase II organizes chromatin topology at a local scale, creating transcription factories (Fig. 6e).

## Discussion

The study of the chromatin architecture of metazoan genomes has uncovered a complex organization combining multiple structural elements, such as chromosome territories, compartments, TADs, and loops [35, 44, 57, 58]. Analyses of several plant species, including Arabidopsis, rice, barley, maize, tomato, sorghum, foxtail millet, and

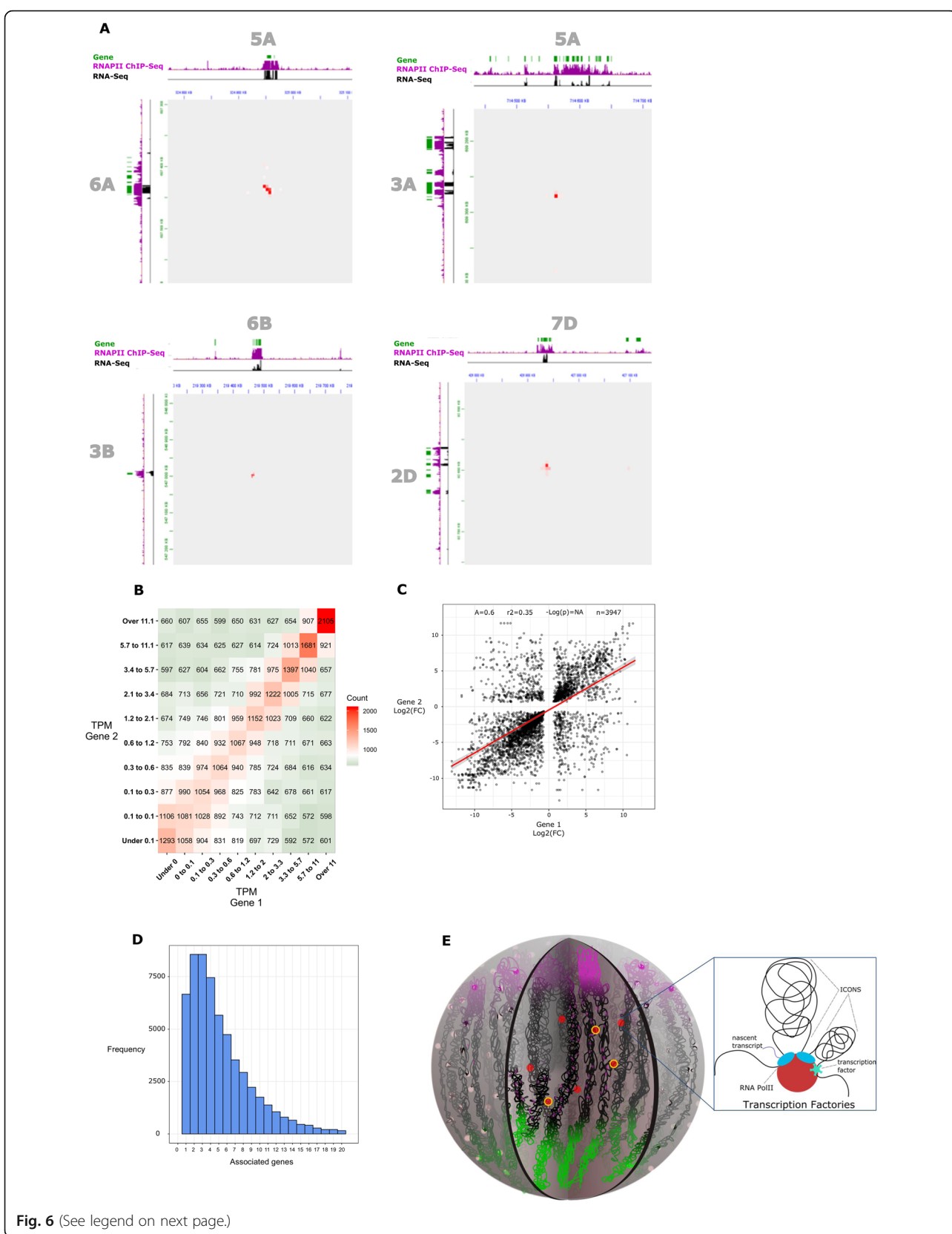

**Fig. 6** (See legend on next page.)

(See figure on previous page.)

**Fig. 6** Gene interchromosomal interaction through transcription factories is associated with coregulation. **a** Integration of RNAPII Hi-ChIP, RNAPII ChIP-seq, and RNA-seq data. Examples of 2D heatmap with the frequency of interchromosomal interactions are presented. Green bars represent genes. **b** Heatmap presenting the gene expression level of gene pairs involved in RNAPII-associated interactions. Within each gene pair, each gene was classified into a decile of expression, and then the number of gene pairs with each combination of deciles was counted. **c** Scatterplot of $\log_2$(shoot/root fold change) for pairs of interacting genes associated through interchromosomal RNAPII-associated contacts. Interactions were identified by an RNAPII Hi-ChIP experiment (see the "Results" section). Only interchromosomal contacts conserved in shoots and roots were used in the analysis. **d** Histogram showing the distribution of genes interacting through RNAPII-associated contacts by numbers of partner genes. **e** Model of a transcription factory. Each factory could contain several RNA polymerase II molecules. These transcription factories could include many factors involved in transcription, such as coactivators, chromatin remodelers, transcription factors, histone modification enzymes, RNPs, RNA helicases, and splicing and processing factors. Multiple genes can be processed by the same factory (two are shown)

cotton, have offered a different scenario that is still poorly understood [14, 15, 59, 60] (see introduction). In this context, the generation and integration of Hi-C, ChIP-seq, Hi-ChIP, and RNA-seq data presented in this study offer a robust insight into the genome topology of hexaploid wheat, an important crop whose exploration has traditionally relied on a cytological approach [61].

The first analysis of the 3D genome organization of a polyploid plant was conducted in cotton and focused mainly on the evolutionary consequences of polyploidization at a local scale [60]. In this study, we show with two genome-wide complementary techniques, such as GISH and Hi-C, that the chromatin of hexaploid wheat is not uniformly distributed across the nucleus but rather occupies the subgenome-specific nuclear compartments. This finding is consistent with previous cytological observations indicating that chromosomes of the same subgenome tend to be physically closer than chromosomes of different subgenomes, regardless of whether they are homeologs [61]. Consequently, we propose that genome territories are the primary level of chromatin spatial organization in this species.

The retention of multiple homeologs following polyploidy promotes the subfunctionalization of genes, increasing the capacity to adapt to different environments. The resulting phenotypic variability from which favorable traits can be selected could have facilitated the domestication of modern wheat.

Despite a considerable degree of synteny among subgenomes due to the phylogenetic relatedness of the three progenitors [30], wheat chromosomes form meiotic pairs only with the respective homolog and never with their homeologs [62]. This mechanism prevents the independent segregation of chromosomes from different genomes and results in disomic inheritance. While the *Ph1* locus has been identified as the major regulator of disomic inheritance and molecularly as a duplicated *ZIP4* gene [63], little is known about the underlying mechanisms which facilitate only homolog pairing rather than homeolog pairing during the telomere bouquet stage at the start of meiosis. The establishment of genome territories may be a mechanism that favors the pairing of homologs versus homeologs by creating territorial "boundaries" between the different subgenomes.

Another important finding is the differential frequency of interaction between specific subgenomes: A and B interact more frequently than do A and D or B and D (Fig. 1b, Additional file 1: Supplemental Figure S1). It is tempting to speculate that this imbalance could be related to the evolutionary history of modern wheat: the hybridization event at the origin of the tetraploid *T. turgidum*, containing the A and B subgenomes, occurred several hundred thousand years before the hybridization between *T. turgidum* and *A. tauschii* (carrying the D genome). As a consequence, the A and B subgenomes have co-existed under the severe spatial constraint dictated by the limited nuclear space, especially considering the large genome size of *T. turgidum* (12 Gbp) [64]), for a longer time than either A or B have co-existed with the recently added D subgenome. Our observation opens the possibility that the A and B subgenomes have established a certain degree of physical interaction before subgenome D was introduced, implying that the chromatin architecture was conserved during the process of allopolyploidization. Accordingly, we propose that the genome territories in hexaploid wheat were inherited from its ancestors and maintained after each hybridization event; however, it cannot be excluded that gradual adjustments occurred during the course of the following evolution.

In hexaploid wheat and several other species, such as barley, rye, oat, *Drosophila melanogaster*, and *Saccharomyces cerevisiae*, chromosomes are arranged in a Rabl configuration, where telomeres and centromeres are localized at opposite sides of the nucleus [65–68]. Consistently, Hi-C interaction maps generated in this study show that chromosomes are V-folded and oriented in the same direction, as apparent from the antidiagonal pattern in the interchromosomal maps (Fig. 1b, f). In addition, we observed a clear polarization of constitutive heterochromatin (marked by H3K27me1 and H3K9me2) and facultative heterochromatin (marked by H3K27me3) within the nucleus, a combined effect of the Rabl configuration and the previously reported subtelomeric enrichment in facultative heterochromatin [34]. On a global scale, the polarization of constitutive and facultative heterochromatin could respond to the necessity to subcompartmentalize genomic regions with different levels of

transcriptional activity and epigenetic marks to localize the enzymatic activities of the transcriptional machinery and epigenetic regulators in a delimited nuclear volume.

In the context of a tightly packed structure imposed by the reduced volume of the nucleus, the architecture of chromatin provides a physical framework for the dynamic and coordinated expression of genes in response to internal and external cues. In metazoans, chromatin is organized in TADs, regions characterized by preferential contacts between the loci inside the same TAD and lack of interaction with the loci in neighboring TADs. These structures are known to facilitate the establishment of enhancer-promoter contacts and to mediate the regulation of gene expression [69, 70]. Here, we found a widespread presence of TAD-like domains in wheat, similar to what has been previously reported in maize, tomato, sorghum, and foxtail [14, 15, 59, 60]. However, the wheat TAD-like formations displayed plant-specific features, such as (i) depletion of genes within the domains; (ii) enrichment in genes, especially active ones, and DNA methylation at their borders; and (iii) propensity to form loops between genes located at the border. To emphasize these peculiar properties, we hence named these domains ICONS, for intergenic condensed spacer.

We hypothesize that ICONS promote the formation of gene-to-gene loops by condensing the intergenic regions, thus bringing into close proximity the genes located at the ICONS borders. This mechanism could also increase the accessibility of genes to the replication machinery and other enzymatic activities, including epigenetic regulators, and bury within ICONS the sequences that should not be accessible, such as repressed genes. Indeed, we detected the presence of genes carrying H3K27me3 within ICONS but a strong depletion of active genes bearing H3K9ac and H3K36me3 (Fig. 3a). In a large genome such as that of hexaploid wheat, reducing the physical distance between genes could be critical for improving the efficiency of nuclear metabolism [5]. Furthermore, we observed that genes forming loops tend to (i) share the same epigenetic marks; (ii) share epigenetic marks with the same physiological significance, i.e., either active or repressive (Fig. 5b); (iii) be expressed at similar levels (Fig. 5c); and (iv) behave concordantly when differentially expressed in two organs (Fig. 5d). This evidence strongly suggests that loops bring into physical proximity genes that have to be targeted by the same epigenetic modifiers and, in the case of active marks, by the transcription machinery. The presence of interchromosomal loops between coregulated genes, as uncovered in this study, indicates that the same mechanism also could be functioning *in trans* at very long distances [36].

In addition, single genes appear to take part in multiple loops (Fig. 6b), producing clusters of loops containing several coregulated genes and possibly enhancers. Such "transcription factories" have been observed previously as subnuclear foci of active RNAPII and regulatory factors, in which several Pol II molecules operate simultaneously [71–73]. We independently confirmed the existence of transcription factories in wheat. One possible advantage of this three-dimensional spatial network is that it facilitates the deployment of transcriptional regulatory factors over a restricted number of "focal points," resulting in a better coordination of gene expression inside the nucleus. This could represent an evolutionarily selected trait that counteracts the disadvantage of large genome size.

## Conclusion

In summary, our findings allow us to propose a model in which chromatin architecture in hexaploid wheat is hierarchically organized around four main topological features: (i) genome territories, (ii) constitutive and facultative heterochromatin polarization, (iii) transcription factories, and (iv) ICONS. This sophisticated organization is required to balance the size and complexity of the hexaploid wheat genome. In a more general perspective, this study highlights that three-dimensional conformation at multiple scales is a key factor governing gene transcription.

## Materials and methods
### Plant material
The wheat cultivar Chinese Spring and the rapeseed cultivar Westar were used for this study. Seeds were surface sterilized with bayrochlore and ethanol, sown on MS agar plates, placed at 4 °C in the dark for 2 days, and then grown in a growth chamber for 14 days (long-day conditions, 18 °C).

### Genomic in situ hybridization of root meristematic cells
The preparation of *T. aestivum* root metaphase cells and subsequent genomic in situ hybridization (GISH) was carried out as described previously [63, 74]. The preparation of root interphase cells was performed as for metaphase cells but omitting the nitrous oxide treatment. *Triticum urartu* and *Aegilops tauschii* were used as probes to label wheat A and wheat D genomes, respectively. Telomere repeat sequence (TRS) probe was amplified by PCR as described previously [75] and labeled with tetramethyl-rhodamine-5-dUTP (Sigma, St. Louis, MO, USA) by nick translation as described previously [76]. *T. urartu* and *Ae. tauschii* genomic DNA were labeled with biotin-16-dUTP and digoxigenin-11-dUTP, using the Biotin-Nick Translation Mix and the DIG-Nick Translation Mix, respectively (Sigma) according to the manufacturer's instructions. Biotin-labeled probes were detected with Streptavidin-Cy5 (Thermo Fisher

Scientific, Waltham, MA, USA). Digoxigenin-labeled probes were detected with anti-digoxigenin-fluorescein Fab fragments (Sigma).

Images were acquired using a Leica DM5500B microscope equipped with a Hamamatsu ORCA-FLASH4.0 camera and controlled by Leica LAS X software v2.0. Images were processed using Fiji (an implementation of ImageJ, a public domain program by W. Rasband available from http://rsb.info.nih.gov/ii/) and Adobe Photoshop CS4 (Adobe Systems Incorporated, USA) version 11.0 x 64.

### Immunofluorescence

Seedlings were fixed in PFA 4% in PHEM (PIPES 60 mM, HEPES 25 mM, EGTA 10 mM, $MgCl_2$ 2 mM pH 6.9) under a vacuum for 20 min. Seedlings were washed for 5 min in PHEM and 5 min in PBS pH 6.9 and chopped on a petri dish in PBS supplemented with 0.1% triton (w/v). The mixture was filtered (50 μm) and centrifugated 10 min at 2000*g*. The supernatant was carefully removed, and the pellet was washed once with PBS, gently resuspended in 20 μl PBS, and a drop was placed on a poly-lysine slide and air-dried. Slides were rehydrated with PBS and permeabilized 2 times by incubating for 10 min in PBST (PBS, 0.1% Tween20 v/v). The slides were placed in a moist chamber and incubated overnight at 4 °C with primary antibody anti-H3K9me2 (Millipore 07-441), anti-H3K27me3 (Millipore, ref. 07-449), anti-H3K27me1 (Abcam ab195492), anti-H3K36me3 (Abcam, ab9050), or anti-polymerase II (Active Motif, 39097) in PBST supplemented with BSA (3% w/v). The slides were washed 5× for 10 min in PBST (at RT) and incubated 1 h at RT in the dark with the secondary antibodies Alexa Fluor 594 goat anti-rabbit (A11037 Invitrogen), Alexa Fluor 488 goat anti-mouse (A11001 Invitrogen), Alexa Fluor 488 goat anti-rabbit (A11034 Invitrogen), or Alexa Fluor 594 goat anti-mouse (A11032 Invitrogen), diluted (1/400 v/v) in PBST, 3% BSA. The slides were washed 5× for 10 min in PBST and mounted with a drop of Vectashield with DAPI and were directly imaged on an upright microscope (Zeiss Microsystems).

### In situ Hi-C assay

Fourteen-day-old shoots or roots were used for in situ Hi-C. The experiment was carried out according to the protocol published by Liu et al. [74] using DpnII enzyme (New England Biolabs) with minor modifications concerning library preparation for which we used NEBNext UltraII DNA library preparation kit (New England Biolabs). For library amplification, nine PCR cycles were performed and Hi-C libraries were purified with SPRI magnetic beads (Beckman Coulter) and eluted in 20 μl of nuclease-free water. The quality of the libraries was assessed with Agilent 2100 Bioanalyzer (Agilent), and the libraries were subjected to 2 × 100 bp paired-end high-throughput sequencing by HiSeq 4000 (Illumina).

### Assay for transposase-accessible chromatin with high-throughput sequencing

One hundred micrograms of 14-day-old seedlings was ground, and the nuclei were isolated with 4 °C using nuclei isolation buffer (0.25 M sucrose, 10 mM Tris-HCl, 10 mM $MgCL_2$, 1% Triton, 5 mM beta-mercaptoethanol) containing proteinase inhibitor cocktail (Roche) and filtered at 63 μm. The nuclei were resuspended in 1× TD buffer (Illumina FC-121-1030) and 2.5 μl of Tn5 transposase (Illumina FC-121-1030) were added. Transposition reaction was performed at 37 °C for 30 min, and DNA was purified using a Qiagen MinElute Kit. DNA libraries were amplified for a total of 8 cycles as described by Buenrostro et al. [49] and Jegu et al. [50].

### RNA-seq assay

Total RNA were extracted from 180 mg of shoots of 12-day-old seedlings with the NucleoSpin RNA kit (Macherey-Nagel), according to the manufacturer's instructions. RNA-seq libraries were prepared using 2 μg of total RNA with the NEBNext® Poly(A) mRNA Magnetic Isolation Module and NEBNext® Ultra™ II Directional RNA Library Prep Kit for Illumina (New England Biolabs), according to manufacturer's recommendations. The quality of the libraries was assessed with Agilent 2100 Bioanalyzer (Agilent).

### RNAPII ChIP assay

RNAPII ChIP assay was performed with anti-RNA polymerase II CTD repeat YSPTSPS antibody (Abcam ab26721) using the same protocol as in IWGSC, 2018 [34].

### Hi-ChIP assay

For Hi-ChIP experiments, the nuclei were isolated from the shoots and roots using the same procedure as in the in situ Hi-C experiments, and the Hi-ChIP protocol from Mumbach et al. 2016 was then applied using the DpnII restriction enzyme (New England Biolabs) and the anti-polymerase II antibody (Active Motif, 39097). The quality of the libraries was assessed with Agilent 2100 Bioanalyzer (Agilent), and the libraries were subjected to 2 × 75 bp paired-end high-throughput sequencing by NextSeq 500 (Illumina).

### Chromosome conformation capture

One gram of 14-day-old shoots was cross-linked in 1% (v/v) formaldehyde at room temperature for 20 min. Cross-linked plant material was ground, and the nuclei were isolated and treated with SDS 0.5% at 62 °C for 5 min. SDS was sequestered with 2% Triton X-100. Digestions were performed overnight at 37 °C using 150 U of DpnII enzyme (New England Biolabs). Restriction enzymes were inactivated by the addition of 1.6% SDS and incubation at 62 °C for 20 min. SDS was sequestered

with 1% Triton X-100. DNA was ligated by incubation at 22 °C for 5 h using 4000 U of T4 DNA ligase (Fermentas). Reverse crosslinking was performed by overnight treatment at 65 °C. DNA was recovered after Proteinase K treatment by phenol/chloroform extraction and ethanol precipitation. Relative interaction frequencies were calculated by quantitative real-time PCR using 15 ng of DNA. A region uncut by DpnII was used to normalize the amount of DNA.

| Primer list for 3Cassays | |
| --- | --- |
| Primer name | Sequence |
| TraesCS1B02G226200 | AAATTGGCTGCCGATTGGTTCG |
| TraesCS1B02G226300 | CTGGGCTAAAAGCCTCACACGTT |
| TraesCS4D02G254100 | TGGCGATAATGCAACATTGCAGAA |
| TraesCS4D02G254200 | ATTTACCTGCAAGGAGAGTTCCCC |
| TraesCS7A02G231500 | AAACAAGCTCTTGAGGTGACATCG |
| TraesCS7A02G231600 | CCCAGTTGGTTATTCGGCAGT |
| TraesCS1D02G176100 | GACTAGCACCCCCAAGCATTCC |
| TraesCS1D02G176200 | AGCTTGGATGCCTGCTTACGG |

## Gene annotation

All the analyses involving genes were carried out using a custom annotation derived from iwgsc_refseqv1.1 (available at https://urgi.versailles.inra.fr/download/iwgsc/IWGSC_RefSeq_Annotations/v1.1/iwgsc_refseqv1.1_genes_2017July06.zip. Briefly, we combined high-confidence genes (file IWGSC_v1.1_HC_20170706.gff3) and a subset of low-confidence genes (file IWGSC_v1.1_LC_20170706.gff3) including only genes overlapping with ChIp-seq peaks of at least one histone mark (H3K9ac, H3K36me3, H3K27me3) (see below). Genes annotated on "unanchored scaffolds" (ChrUn) were removed. The final annotation (file IWGSC_v1.1.noChrUn.gene. HC+LC_over_peaks.gff3) contained 165,277 genes of which 105,088 were high-confidence and 60,189 were low-confidence.

## Identification of ncRNAs

Transcript classes were predicted by assessing the coding potential of each gene with Coding Potential Calculator (CPC) [77] and its improved version CPC2 [78], both with default parameters and by comparison with the RFAM database 13.0 [79] using cmscan of the suite infernal v1.1.1 [80]. In case of a hit corresponding to structural RNA (tRNA, rRNA, snRNA, or snoRNA), the transcript was classified as "structural"; then, if CPC and CPC2 both predicted the transcript as non-coding, we classified it as "non-coding," otherwise as "coding."

## Analysis of Hi-C data

Raw FASTQ files were preprocessed with Trimmomatic v0.36 [81] to remove Illumina sequencing adapters. 5′ and 3′ ends with a quality score below 5 (Phred+33) were trimmed, and reads shorter than 20 bp after trimming were discarded. The command line used is "trimmomatic-0.36 PE -phred33 -validatePairs ILLUMINACLIP:TruSeq3- SE.fa:2: 30:10 LEADING:5 TRAILING:5 MINLEN:20". Trimmed FASTQ files were processed with the pipeline HiC-Pro v2.9.0 [82] with minor modifications. Briefly, reads were aligned to iwgsc_refseq1.0 reference genome assembly (available at https://urgi.versailles.inra.fr/download/iwgsc/IWGSC_RefSeq_Assemblies/v1.0/iwgsc_refseqv1.0_all_chromosomes.zip. using bowtie2 v2.3.3 [83] with default settings, except for the parameter "--score-min L, -0.6, -0.8". Forward and reverse mapped reads were paired and assigned to DpnII restriction fragments. Invalid ligation products (such as dangling ends, fragments ligated on themselves, and ligations of juxtaposed fragments) were discarded.

A summary of Hi-C mapping statistics is provided in Additional file 4, Table S3. Valid pairs were used to produce raw interaction matrixes at various resolutions. Normalized matrixes were produced by iterative correction using the "ice" utility provided as part of HiC-Pro [84]. Pre-computed .hic files were generated with the software Juicer Tools and visualized with the tool Juicebox v1.9.8 [85]. Interaction frequency boxplots (Fig. 1c) where created using the packages HiTC v1.26.0 [86] and ggplot2 v3.1.0 [87] in R v3.4.3 statistical environment [88]. Single-chromosome Hi-C interaction heatmaps and H3K27me3 read density tracks (Fig. 1f and Additional file 1: Supplemental Figure S4) were produced with HiCPlotter v0.8.1 at 1-Mb resolution [89].

## Insulation index and ICONS

We identified ICONs with the software TADtool v0.77 [90] using the "insulation index" method [48]. The "insulation index" of a genomic region is defined as the normalized average frequency of interaction with the neighboring sequences within a pre-defined distance. Regions with an insulation index above a pre-established threshold are identified as folding domains.

In this study, we used ice-normalized matrixes at 25 kb of resolution with a scoring window of 100 kb and a calling threshold of 0.4. The histogram of ICONS distribution by size (Additional file 1: Supplemental Figure S5) was plotted with the package ggplot2 v3.1.0 [87] and R. For Fig. 3d-f, j, the genome was partitioned in 10 deciles of 25-kb bins according to their insulation index using the function *mutate* of the package dplyr v0.7.8 in R. Groups of genes lying in each genomic partition of insulation

index (see above) were identified using the bedtools v2.27.1 [91] *intersect* with the command "for i in {1..10}; do bedtools intersect -sorted -wb -F 0.5 -a DECILE.$i.bed -b IWGSC_v1. 1.noChrUn.gene. HC+LC_over_peaks.gff3 > IWGSC_v1.1 .DECILE.$i.gff3; done".

### Analyses of ChIP data
We used the histone marks ChIP-seq datasets from IWGSC, 2018 (NCBI SRA accession SRP126222). Histone marks and RNAPII ChIP were pre-processed with Trimmomatic v0.36 [81] to remove Illumina sequencing adapters. 5′ and 3′ ends with a quality score below 5 (Phred+33) were trimmed, and reads shorter than 20 bp after trimming were discarded. The command used is "trimmomatic-0.36.jar PE -phred33 - validatePairs ILLU-MINACLIP:TruSeq3-PE.fa:2:30:10 LEADING:5 TRAIL-ING:5 MINLEN:20". Trimmed paired ends reads were aligned against the reference genome iwgsc_refseq1.0 using bowtie2 v2.3.3 with the setting "--very-sensitive". Alignments with MAPQ < 10 and duplicated reads were discarded with sambamba v0.6.8.

Normalized short-read density was calculated using the utility bamCoverage from the bioinformatics suite deeptools2 3.1.0 [92] with the command line "bamCoverage -p 4 -bs 100 --extendReads -of bigwig --normalizeTo1x 14547261565 -b INPUT.bam -o OUTPUT.bigwig". Peaks of Chip-seq read density were identified using the software MACS2 [93] with the command "macs2 callpeak -f BAM --nomodel --to-large -p 0.01 -broad -g 17e9 --bw 300" for histone marks and "macs2 callpeak -f BAMPE --nomodel −q 0.001 --broad-cutoff 0.01 -g 17e9 --bw 300" for RNA-PII. Histone mark peaks with a fold change lower than 3 were discarded.

### Analyses of ATAC-seq data
ATAC-seq paired ends reads were aligned against the reference genome iwgsc_refseq1.0 using bowtie2 v2.3.3 with the setting "--very-sensitive" [83]. Alignments with MAPQ < 10 were filtered with sambamba v0.6.8. Short-read density was calculated using bamCoverage (see above) with the command "bamCoverage -bs 10 -- centerReads -b INPUT.BAM -of bigwig -o OUTPUT.bigwig". Peaks of the read density were identified with the command "macs2 callpeak -f BAMPE --nomodel -q 0.001 --broad- cut-off 0.01 -g 17e9 --bw 300".

### Analyses of RNA-seq data
Transcript abundance in shoots was quantified directly from the fastq files with kallisto [94] and converted to gene-level counts (TPM) with the R package tximport [95]. Genes were classified as expressed or non-

expressed and split in quantiles of expression with R using the median value of the three biological replicates.

### Differential expression analysis
RNA-seq data relative to seedling shoot and root were obtained from the dataset published by Oono et al. [55]. Paired short reads were preprocessed like ChIP-seq data (see above) to remove Illumina sequencing adapters. Trimmed paired-end reads were aligned against the reference genome iwgsc_refseq1.0 using STAR_2.5.4b [96] with the command "STAR --genomeDir /STAR_iwgsc1.0 --sjdbGTFfile /IWGSC_v1.1_HC+LC_20170706.gtf --sjdbOverhang 100 --outSAMstrandField intronMotif --quantMode GeneCounts --outSAMtype BAM SortedByCoordinate -- outFilterIntronMotifs RemoveNoncanonical". Genes differentially expressed ($p$ value < 0.01) were identified from raw read counts with DESeq2 v1.22.1 in R.

### Generation of meta-plots
Meta-gene profiles of insulation index and the meta-ICONs profiles of epigenetic and genomic features (histone marks, RNAPII, ATAC-seq, DNA methylation, TE coverage) were generated with deepTools2 v3.1.0 [92] *computeMatrix* and plotted with *plotProfile.* Shuffled ICONS and genes were generated with bedtools *shuffle.*

Genes overlapping or non-overlapping with ChIP peaks of each histone mark and RNAPII (Fig. 3a–c, g; Additional file 1: Supplemental Figure S1 I and S7 B-C and F-G) were identified using the tool *intersect* of the bioinformatics suite bedtools v2.27.1 with the command: "bedtools intersect -sorted -wa -a IWGSC_v1.1.noChrUn.gene. HC+LC_over_peaks.gff3 -b PEAK_FILE | sort -k4,4 -u > OVER-LAPPING_GENES.GFF3" and "bedtools intersect - sorted -v -wa -a IWGSC_v1.1.noChrUn.gene. HC+LC_over_peaks.gff3 -b PEAK_FILE | sort -k4,4 -u > NONOVER-LAPPING_GENES.GFF3". For ATAC-seq, only genes containing a peak within 500-bp upstream of TSS were classified as overlapping.

DNA methylation data were obtained from Ramirez-Gonzalez et al. [36] and are available in the NCBI SRA repository under accession code SRP133674. Genomic coverage of transposable elements was calculated using the annotation available at https://urgi.versailles.inra.fr/download/iwgsc/IWGSC_RefSeq_Annotations/v1.0/iwgsc_refseqv1.0_TransposableElements_2017Mar13.gff3.zip.

### Identification and analysis of genomic interactions
Gene-to-gene loops (GGLs), RNAPII-associated loops (RALs), and interchromosomal interactions were identified using HOMER v4.10 [93] at a resolution of 20 kb, 25 kb, and 50 kb, respectively, with $p$ value < 0.05, $Z$-score > 1.5, and FDR < 0.1.

For GGLs, the first and second genomic bins of shoot interactions were separately annotated with genes and then with ChIP peaks and TPMs using bedtools intersect. Bins not overlapping with genes were discarded, and the first and second bins were rejoined interaction-wise with the function *merge* in R. We then transformed the annotated gene pairs into a database reporting, for each gene, the presence ("1") or absence ("0") of histone marks or RNAPII, a value of TPM; the expression status ("expressed" or "non-expressed"); plus the distance between the two genes and the $Z$-score calculated by HOMER.

Starting from the database, we counted the number of genes with and without each feature and reported the aggregated scores in a series of $2 \times 2$ contingency tables with the counts for the first gene in rows and for the second gene in columns. We explored all possible feature combinations and applied a $X^2$ test to each $2 \times 2$ table. Since in a large population such as the one examined here, a statistically significant $p$ value can arise from small differences between observed and expected values [97], we reported the $\log_2$ of odds ratios to interpret the outcome of the test for each combination of marks (Additional file 3: Table S2) and calculated Cramer's $V$, a measure of the "effect size." GGLs between expressed genes (TPM > 0) were split in two series of quartiles according to the TPMs of the two genes, and the counts in each combination of quartiles are indicated in the heatmap in Fig. 4c. The same procedure was followed to produce Fig. 6b by splitting interchromosomal RNAPII-associated interactions in ten quantiles of TPM.

Finally, we selected GGLs, RALs (intrachromosomal), and interchromosomal RNAPII-associated interactions conserved in both shoots and roots. Using bedtools intersect, we annotated with differentially expressed genes the first and the second genomic bins of each interaction. Bins not overlapping with genes were discarded, and the first and second bins were rejoined interaction-wise with the function *merge* in R. The linear regression between the $\log_2$(fold change shoot/root) of the two genes of all pairs was calculated with R, and scatter plots (Figs. 4d, 5a, and 6c) were generated with ggplot2 v3.1.0. Quartiles of the distance between the two genes (Additional file 1: Supplemental Figure S10) and quartiles of $Z$-score in shoots and roots (Fig. 4e, Additional file 1: Supplemental Figure S11 and S14) were calculated using the function *mutate* of the R package dplyr v0.7.8 [98], and both regression and scatterplot reproduced for each group.

We counted the number of partners of each gene in interchromosomal RNAPII-associated interactions with the R function *table* and plotted the histogram in Fig. 6d with ggplot2.

## Supplementary information

---

**Additional file 1.** Supplementary figures.

**Additional file 2: Table S1**. with information about gene to gene loops (GGLs): position, size, strength, and presence or absence of H3K9ac, H3K27me3, H3K36me3 histone marks, presence or absence of RNA polymerase II, transcriptional status and expression level (TPM).

**Additional file 3: Table S2**. With odds ratios (OR) and effect size (Cramer's V) for all combinations of features of interacting genes pairs.

**Additional file 4: Table S3**. With a summary of Hi-C and HiChIP libraries statistics.

**Additional file 5:** Review history.

---

**Peer review information**

**Review history**

The review history is available as Additional file 5.

**Authors' contributions**

LC, DL, and MB designed the research. DL, NRG, YH, and MP collected the plant material and performed the Hi-C, Hi-ChIP, ChIP-seq, and RNA-seq experiments. SK prepared the Hi-C libraries. DL, SK, and MP performed the Illumina sequencing. AMR, SDuncan, and SDomenichini contributed to the microscopy, the GISH experiment, and the immune-localization experiment. LC, TB, CP, DMM, and AV performed the bioinformatics analyses. LC, JSRP, and MB wrote the manuscript, with support from all authors. EP, SD, GM, HH, CB, MC, MMM, AB, CL, AH, CR, DL, and MB analyzed the data and provided critical feedback. The authors read and approved the final manuscript.

**Funding**

This work was funded by the Agence National de la Recherche ANR (3DWheat project ANR-19-CE20-0001-01) and by the Institut Universitaire de France (IUF).

**Availability of data and materials**

Sequencing data from wheat Hi-C, Hi-ChIP, ATAC-seq, and RNA-seq and RNA polymerase II ChIP-seq, and rapeseed Hi-C are stored and openly available at the GEO under accession GSE133885 [99].
DNA methylation data used for Additional file 1: Supplemental Figure S6 were obtained from Ramirez-Gonzalez et al. [36] and are available in the NCBI SRA repository under accession code SRP133674.
RNA-seq data from the seedling shoots and roots used for the differential expression analysis were generated by a previous study [55] and are available in the NCBI SRA repository under accession code DRR003148, DRR003149, and DRR003150 (seedling root) and DRR003154, DRR003155, and DRR003156 (seedling shoot).

**Ethics approval and consent to participate**

Not applicable

**Competing interests**

The authors declare that they have no competing interest

**Author details**

[1]Institute of Plant Sciences Paris of Saclay (IPS2), UMR 9213/UMR1403, CNRS, INRA, Orsay, France. [2]Division of Biological and Environmental Sciences and Engineering, King Abdullah University of Science and Technology, Thuwal 23955-6900, Kingdom of Saudi Arabia. [3]John Innes Centre, Norwich Research Park, Norwich NR4 7UH, UK. [4]Earlham Institute, Norwich Research Park, Norwich NR4 7UG, UK. [5]INRA UMR1095 Genetics, Diversity and Ecophysiology of Cereals, 5 chemin de Beaulieu, 63039 Clermont-Ferrand, France. [6]Center for Plant Molecular Biology (ZMBP), University of Tübingen, Auf der Morgenstelle 32, 72076 Tübingen, Germany. [7]Institut Universitaire de France (IUF), Paris, France.

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

## 

