## [**Additional file 5:** Review history. · Genome Biology]

Review History

First round of review

Reviewer 1

Are you able to assess all statistics in the manuscript, including the appropriateness of statistical tests used? Yes, and I have assessed the statistics in my report.

Comments to author:

This is a complex paper and interesting paper. It utilized numerous approaches, such as ChIP-seq, ATAC-seq, Hi-C and Hi-ChIP and RNAPII antibody data, to study the organization of the wheat nucleus and spacial factors affecting transcription. The methods are appropriate and well executed. Conclusions are usually statistically supported. The analyses confirmed compartmentalization of the wheat subgenomes, which has been reported earlier. In addition, the study characterized spacial interactions between transcriptionally active genes and euchromatic histone modifications. The paper shows relationships between epigenetic marks, chromosome conformation, and 3D organization of gene expression in the wheat nucleus and presents a model of gene transcription in polyploid wheat. The paper contains a large amount of data and numerous technique. As such, it is not easy to read. Results would benefit if each section would start with a hypothesis or explanation of what is expected and what that specific analysis means rather than relying on the reader to go to M&M or publications to get the background.

Specific comments:

Line 138 Homoeologues is misspelled.

Lines 152. Remove "interestingly". It confuses the flow of text. The meaning of labelling with H3K27me1 H3K27me3 needs to be described more clearly.

Line 190. "Unexpected" ...What is expected? What are noncoding genes? Please define.

Line 193 What these histone marks are expected to indicate?

Line 230 Should be: "RNA polymerase II ChIP-seq"

Line 248 TPM has already been defined.

Line 249 Staring from the database... Not clear. Delete?

Line 269 Define differentially expressed.

Line 304 Define that RNAPII is RNA polymerase II-mediated...

Lines 327-376.subfunctionalization of genes on homoeologous chromosomes has enabled wheat domestication" There is no evidence in support of this statement. Wheat (meaning tetraploid wheat) was domesticated for morphological reasons (large and nutritious seeds). There are many diploid domesticates, e.g., barley. I suggest to delete this argument.

Page 8 Abbreviations such as CTCF, ICE are not explained.

References need editing. They have been downloaded without being checked.

Figure 1A is not good enough. It is stated in Introduction that the source of the B genome is unknown. Yet, there is a picture of it. Is not that contradictory?

Fig. 1F need labeling on vertical axis and legend needs more explanation

Reviewer 2

Are you able to assess all statistics in the manuscript, including the appropriateness of statistical tests used? Yes, and I have assessed the statistics in my report.

Comments to author:

The manuscript by Concia et al reported Hi-C and RNAP2 HiChIP data analyses in wheat seedlings and root. they are trying to address a very important and interesting topic. I believe these are wonderful pieces of data and will be very useful to the field. However, the current presentation of the results is insufficient and it is difficult to gauge the findings and interpretations due to lack of clarity on the specific analyses.

1. I can't find a detailed data quality summary about HiChIP and Hi-C data to tell whether the data is good or not. At least the authors should provide a comprehensive tables with the total # of reads, the # of valid reads etc. used for downstream analysis.

2. The authors observe more frequent interactions between subgenome A and B than A/D or B/D, and I think it is Okay to say these results indicate the genetic proximity between subgenomes, however, it is far fetched to say it suggesting the order of hybridization between the three ancestors during evolutionthey.

3. The observation of opposite and polarized distribution of facultative and constitutive heterochromatin marks throughout the nucleus is quite interesting. How many immunostained nuclei examined and what is % of nuclei showing the observed distribution? In addition, How does the authors explain that the second strong interaction signal observed in contacts heatmaps are out of antidiagonal line in only one direction in most of chromosomes?

4. The authors observed a lot of high frequent chromatin interactions, and I would say you'd better validate some of them using FISH.

5. RNA-seq assays were performed using 12-d-old shoot and ATAC-seq/in situ Hi-C data are from 14-d-old shoot. what's stage of the tissue used for HiChIP? why don't you choose the exact same stage of tissue for those high-throughput assays when you design the project?

6. The authors generate the HiChIP datasets, but I can not find any detail and description about its quality. Moreover, taking advantages of HiChIP's high-resolution contact maps, You should extend the analyses to promoter-promoter and enhancer-promoter loops in wheat.

We would like to thank both reviewers for their excellent comments. We have fully addressed all the concerns and comments raised by both reviewers. These comments have helped us to significantly improve the manuscript.

We highlighted in yellow all the sentences added in the new version of the manuscript.

Point-by-point Response to Reviewers:

Reviewer reports:

Reviewer #1: This is a complex paper and interesting paper. It utilized numerous approaches, such as ChIP-seq, ATAC-seq, Hi-C and Hi-ChIP and RNAPII antibody data, to study the organization of the wheat nucleus and spacial factors affecting transcription. The methods are appropriate and well executed. Conclusions are usually statistically supported. The analyses confirmed compartmentalization of the wheat subgenomes, which has been reported earlier. In addition, the study characterized spacial interactions between transcriptionally active genes and euchromatic histone modifications. The paper shows relationships between epigenetic marks, chromosome conformation, and 3D organization of gene expression in the wheat nucleus and presents a model of gene transcription in polyploid wheat. The paper contains a large amount of data and numerous technique. As such, it is not easy to read. Results would benefit if each section would start with a hypothesis or explanation of what is expected and what that specific analysis means rather than relying on the reader to go to M&M or publications to get the background.

Specific comments:

Line 138, now line 147. Homoeologues is misspelled.

R/We thank the Reviewer for pointing out the misspelling. We corrected it throughout the paragraph.

Lines 152, now line 168. Remove "interestingly". It confuses the flow of text. The meaning of labelling with H3K27me1 H3K27me3 needs to be described more clearly.

R/We removed "interestingly" as requested, by replacing the sentence "*Interestingly, we observed an opposite and polarized distribution of these two marks throughout the nucleus, which suggests the presence of subnuclear domains with different types of chromatin*" with "*We observed an opposite and polarized distribution of these two marks throughout the nucleus, which suggests the presence of subnuclear domains with different types of chromatin.*"

We added the following sentence to describe the epigenetic role of H3K27me1 and H3K27me3: "*H3K27me3 is plays a critical role in the epigenetic silencing of developmentally or stress-regulated genes in the context of facultative (reversible) heterochromatin, while H3K27me1 is required for the formation and maintenance of constitutive heterochromatin, and thus participates in the inhibition of TE expression (Jacob and Michaels 2009; Grob et al. 2014)*"

Line 190, now line 213. "Unexpected" ...What is expected? What are noncoding genes? Please define.

R/We deleted the word "unexpected" and added a more precise definition of coding gene and noncoding genes.

The sentence "*Unexpectedly, both protein-coding genes (PCGs) and non-coding genes tended to localize over local minima of the insulation index and in particular at the boundaries of folding domains (Fig. 2, B to E)*" was rephrased into "*A prominent feature of the insulation index was the presence of genes, both protein-coding and non-coding such as microRNA precursors or long noncoding*"

RNA, in correspondence of local minima and in particular at the boundaries of folding domains (Fig. 2, B to E)''

Line 193, now line 217. What these histone marks are expected to indicate?

R/We added a reference to the physiological meaning of these histone marks by replacing "Examining the distribution of H3K9ac, H3K36me3, and H3K27me3 over ICONS" with "Examining the distribution of the active chromatin marks H3K9ac, H3K36me3, and of the facultative heterochromatin mark H3K27me3 over ICONS"

Line 230, now line 254 Should be: "RNA polymerase II ChIP-seq".

R/We edited accordingly.

Line 248, now line 279 TPM has already been defined.

R/The definition of TPM has been deleted.

Line 249, now line 276 Starting from the database... Not clear. Delete?

R/We replaced "To facilitate the analysis, we compiled a database" with "To facilitate the analysis, we compiled a table (Table S1)" at line 276.

For consistency, the sentence "Starting from the database, we counted the frequencies of each feature on genes involved in GGLs and crossed the frequencies of the first and the second gene" was replaced with "We counted the frequencies of each feature on genes involved in GGLs and crossed the frequencies of the first and the second gene in Table S1". (line 282)

We show here a subset of Table S1 for clarity

Gene_1	Gene_1 strand	Gene_1 TPM	Gene_1 H3K27me3	Gene_1 H3K36me3	Gene_1 H3K4me3	Gene_1 H3K9ac	Gene_1 RNAPII	Gene_1 Expression	Gene_2	Gene_2 strand	Gene_2 TPM	Gene_2 H3K27me3	Gene_2 H3K36me3	Gene_2 H3K4me3	Gene_2 H3K9ac	Gene_2 RNAPII	Gene_2 Expression
TraesCS1A02C	+	-	1	0	0	0	1	0	TraesCS1A02C	+	-	1	0	1	0	0	0
TraesCS1A02C	-	0,10	1	0	1	0	0	1	TraesCS1A02C	-	-	0	0	0	0	0	0
TraesCS1A02C	-	0,52	1	0	0	0	1	1	TraesCS1A02C	+	-	1	0	0	0	0	0
TraesCS1A02C	+	-	1	0	0	0	1	0	TraesCS1A02C	+	-	0	0	0	0	0	0
TraesCS1A02C	+	-	0	0	0	0	0	0	TraesCS1A02C	+	11,06	0	1	1	1	1	1

Line 269, now line 304 Define differentially expressed.

R/We replaced "We subsequently wanted to probe whether gene pairs connected in GGLs show similar changes in expression levels when differentially expressed" with "We subsequently wanted to probe whether gene pairs connected in GGLs show similar changes in expression levels between shoots and roots".

A more accurate definition is given in the following lines (304-307): "Using publicly available gene expression data (Oono et al. 2013), we identified 21,796 differentially expressed (DE) genes between wheat seedling shoots and roots (p-value < 0.01) (see Materials and Methods)."

Line 304, now line 341 Define that RNAPII is RNA polymerase II-mediated...

R/We edited "RNAPII" to "RNA polymerase II" in the title, in this paragraph and in the following.

Lines 327-376, now lines 413 to 417subfunctionalization of genes on homoeologous chromosomes has enabled wheat domestication" There is no evidence in support of this statement. Wheat (meaning tetraploid wheat) was domesticated for morphological reasons (large and nutritious seeds). There are many diploid domesticates, e.g., barley. I suggest to delete this argument.

R/We replaced the sentence "The retention of multiple homoeologues following polyploidy is crucial for subfunctionalization of genes, and therefore a major driver of genomic plasticity. This,

in turn, allows adaptation to different environments and creates the phenotypic variability from which favorable traits can be selected, a process that enabled the domestication of modern wheat” with “The retention of multiple homeologues following polyploidy promotes the subfunctionalization of genes, increasing the capacity to adapt to different environments. The resulting phenotypic variability from which favorable traits can be selected, could have facilitated the domestication of modern wheat.”

Page 8 Abbreviations such as CTCF, ICE are not explained.

R/We address this point by providing a full definition of ICE at lines 207 and describing the function of CTCF as transcriptional repressor at line 201.

References need editing. They have been downloaded without being checked.

R/We carefully checked the references and edited inaccuracies

Figure 1A is not good enough. It is stated in Introduction that the source of the B genome is unknown. Yet, there is a picture of it. Is not that contradictory?

R/As the Reviewer correctly pointed out, the origin of B genome is still unknown. After reviewing the most recent literature we found that the B could have a polyphyletic origins and *Aegilops speltoides* is among the progenitors (Zhang et al. 2018). We therefore edited the Introduction (lines 100 to 105) and modified figure 1A to convey this uncertainty.

Fig. 1F need labeling on vertical axis and legend needs more explanation

R/We added the labeling on vertical axis of Figure 1 F and further elaborate the relevant legend, adding “The heatmap of intra-chromosomal interaction frequency of chromosome 4 B (in red, upper panel) is presented together with the read density of H3K27me3 ChIP-Seq (in green, lower panel). The genomic coordinates are indicated on the left side of the heatmap for Hi-C and under the read density plot for H3K27me3. The colorbar on the right side of the heatmap shows the interaction frequency scale.”

Reviewer #2: The manuscript by Concia et al reported Hi-C and RNAP2 HiChIP data analyses in wheat seedlings and root. They are trying to address a very important and interesting topic. I believe these are wonderful pieces of data and will be very useful to the field. However, the current presentation of the results is insufficient and it is difficult to gauge the findings and interpretations due to lack of clarity on the specific analyses.

1. I can't find a detailed data quality summary about HiChIP and Hi-C data to tell whether the data is good or not. At least the authors should provide a comprehensive table with the total # of reads, the # of valid reads etc. used for downstream analysis.

R/We fully agree with the Reviewer about the importance of releasing these numbers to evaluate the quality of the libraries. We provide a summary of HiC data in Supplemental Table S3.

Supplemental Table S3

	Sequenced reads	Mapped reads	ValidPairs	ValidPairs after removing duplicates
HiC shoot	1,609,463,910	1,482,226,815	1,153,483,099	1,120,241,836
HiC root	2,340,438,410	1,121,631,973	774,100,729	730,708,660
HiChIP shoot	603,633,538	510,619,694	314,477,132	258,021,720
HiChIP root	371,713,047	208,581,121	129,496,024	123,858,326

2. The authors observe more frequent interactions between subgenome A and B than A/D or B/D, and I think it is Okay to say these results indicate the genetic proximity between subgenomes, however, it is far-fetched to say it suggesting the order of hybridization between the three ancestors during evolution. **R/We removed the sentence indicated from the Results section (line 143).**

In addition we formulated this hypothesis in a more speculative way in the Discussion section (lines from 413 to 417). We replaced the sentence *"The retention of multiple homoeologues following polyploidy is crucial for subfunctionalization of genes, and therefore a major driver of genomic plasticity. This, in turn, allows adaptation to different environments a 374 nd creates the phenotypic variability from which favorable traits can be selected, a process that enabled the domestication of modern wheat"* with *"The retention of multiple homeologues following polyploidy promotes the subfunctionalization of genes, increasing the capacity to adapt to different environments. The resulting phenotypic variability from which favorable traits can be selected, could have facilitated the domestication of modern wheat"*

3. The observation of opposite and polarized distribution of facultative and constitutive heterochromatin marks throughout the nucleus is quite interesting. How many immunostained nuclei examined and what is % of nuclei showing the observed distribution?

R/We observed this polarization in 100% (n=50) of the observed nuclei.

In addition, how do the authors explain that the second strong interaction signal observed in contacts heatmaps are out of antidiagonal line in only one direction in most of chromosomes?

R/We welcome the opportunity to clarify better this point. Taking as reference the antidiagonal line passing by the centromere, the interaction signal is always curved towards the longest arm. The reason is that in the Rab1 configuration the telomeres of each chromosome are in physical proximity and the longest arm is bent to accommodate this arrangement and establishes occasional contacts along its whole length. This is particularly evident when the difference of size between the two arms is very high (figure A) but almost not visible when both arms have the same size (figure B). The same

effect can be detected in other plants with Rab1 nuclear configuration such as *Hordeum vulgare* (Mascher et al. 2017) and *Zea mays* (Dong et al. 2017).

To conclude, the second strong interaction signal observed in contacts heatmaps are out of antidiagonal line in only one direction in most chromosomes because we represented all the chromosome with the shortest arm in the leftmost/upper part of the interaction maps.

2D heatmap with the frequency of intrachromosomal interactions for chromosomes 1A (A) and 7A (B). The blue and green bars represent the short and long arms, respectively. The centromeres position is indicated by the light blue dot. The green dashed lines are the diagonals passing by the centromere.

The anti-diagonal signal is bent towards the long arm when the two arms have different size (A) and not when they are similar (B)

4. The authors observed a lot of high frequent chromatin interactions, and I would say you'd better validate some of them using FISH.

Because doing FISH on non-mitotic chromosome for single gene in the hexaploid wheat is very challenging, instead of FISH we validated four interactions using 3C-qPCR. For that end, we selected four gene pairs spanning a distance between 200-kb to 400-kb and performed two biological replicates of 3C-qPCR assay with and without cross linking. These data are presented now in Supplemental Figure S8.

Supplemental Figure S8. Validation of gene to gene loop interaction by chromosome conformation capture assay followed by quantitative PCR (3C-QPCR).

Right: 2D heatmap with the frequency of intrachromosomal interactions are presented for four gene to gene loop interactions, namely *TraesCS1B02G226200* - *TraesCS1B02G226300*; *TraesCS4D02G254100* - *TraesCS4D02G254200*; *TraesCS7A02G231500* - *TraesCS7A02G231600*; *TraesCS1D02G176100* - *TraesCS1D02G176200*. Circular arc highlights the interaction found by Hi-C analysis. Green bars represent genes. The dashed blue circle highlights the interaction in the 2D heatmap.

Left: Schematic representation of the interacting loci. The position of each primer pair used for 3C-qPCR is indicated by a grey arrow. The black arrow indicates the position of the transcription start site. The dash grey line indicates the position of the DpnII restriction site for each locus. The dash blue line pointed the tested interaction. Relative interaction frequencies were calculated as described in Materials and Methods. Two biological replicates are presented.

5. RNA-seq assays were performed using 12-d-old shoot and ATAC-seq/in situ Hi-C data are from 14-d-old shoot. what's stage of the tissue used for HiChIP? why don't you choose the exact same stage of tissue for those high-throughput assays when you design the project?
R/We thank the Reviewer for noticing this typographical error. After verifying the experimental notes we can confirm that all the assays were conducted on 14-d-old shoots and roots. The Methods section has been edited accordingly.

6. The authors generate the HiChIP datasets, but I cannot find any detail and description about its quality. Moreover, taking advantages of HiChIP's high-resolution contact maps, You should extend the analyses to promoter-promoter and enhancer-promoter loops in wheat.

R/We provide a summary of HiChIP data in Supplemental Table 3, as for comment 1.

In addition, we show that the read density of HiChIP “dangling ends” and the read density of RNA polymerase II ChIP-Seq over genes have very similar profiles (Supplemental Figure S11 C). The “dangling ends” occur when forward and reverse reads of a pair are aligned on the same restriction fragment, therefore representing a regular ChIP experiment.

We agree with the Reviewer that extending the analysis to the enhancers-promoter and promoter-promoter loops would provide useful insights in the relation between chromatin architecture and gene regulation. We therefore attempted to identify significant interactions at 1-kb and 5-kb of resolution. Unfortunately we could not find statistically significant interactions. We believe that at the available sequencing depth for such small genomic bins is too low to obtain a sufficient number of reads to provide statistical significance.

Dong P, Tu X, Chu PY, Lü P, Zhu N, Grierson D, et al. Comprehensive mapping of long range interactions reveals folding principles of the human genome. *Mol Plant* [Internet]. 2017;10(12):1497–509. Available from: <https://doi.org/10.1016/j.molp.2017.11.005>

Grob S, Schmid MW, Grossniklaus U. Hi-C Analysis in Arabidopsis Identifies the KNOT, a Structure with Similarities to the flamenco Locus of Drosophila. *Mol Cell* [Internet]. 2014 Jun;55(5):678–93. Available from: <http://dx.doi.org/10.1016/j.molcel.2014.07.009>

Jacob Y, Michaels SD. H3K27me1 is E(z) in animals, but not in plants. *Epigenetics* [Internet]. 2009/08/03. 2009 Aug 16;4(6):366–9. Available from: <https://www.ncbi.nlm.nih.gov/pubmed/19736521>

Mascher M, Gundlach H, Himmelbach A, Beier S, Twardziok SO, Wicker T, et al. A chromosome conformation capture ordered sequence of the barley genome. *Nature* [Internet]. 2017 Apr;544(7651):427–33. Available from: <http://dx.doi.org/10.1038/nature22043>

Oono Y, Kobayashi F, Kawahara Y, Yazawa T, Handa H, Itoh T, et al. Characterisation of the wheat (*triticum aestivum* L.) transcriptome by de novo assembly for the discovery of phosphate starvation-responsive genes: gene expression in Pi-stressed wheat. *BMC Genomics* [Internet]. 2013;14(1):77. Available from: <http://bmcbgenomics.biomedcentral.com/articles/10.1186/1471-2164-14-77>

Zhang W, Zhang M, Zhu X, Cao Y, Sun Q, Ma G, et al. Molecular cytogenetic and genomic analyses reveal new insights into the origin of the wheat B genome. *Theor Appl Genet*. 2018 Feb;131(2):365–75.

Second round of review

Reviewer 2

The revised manuscript is now much better. but I still have a couple of questions.

1. Although the authors provide the details of Hi-C and HiChIP data, but the # of libraries/replicates and the reproducibility between them are still missing.
2. I see a lot of format issues of citations in the M&M section that need to be corrected, e.g. line 528.